# Infant microbiome cultivation and metagenomic analysis reveal *Bifidobacterium* 2'-fucosyllactose utilization can be facilitated by coexisting species

Yue Clare Lou [1,2], Benjamin E. Rubin[2], Marie C. Schoelmerich [2,8], Kaden S. DiMarco[2], Adair L. Borges [2], Rachel Rovinsky[2,9], Leo Song [2], Jennifer A. Doudna [2,3,4,5] & Jillian F. Banfield [2,6,7]

The early-life gut microbiome development has long-term health impacts and can be influenced by factors such as infant diet. Human milk oligosaccharides (HMOs), an essential component of breast milk that can only be metabolized by some beneficial gut microorganisms, ensure proper gut microbiome establishment and infant development. However, how HMOs are metabolized by gut microbiomes is not fully elucidated. Isolate studies have revealed the genetic basis for HMO metabolism, but they exclude the possibility of HMO assimilation via synergistic interactions involving multiple organisms. Here, we investigate microbiome responses to 2'-fucosyllactose (2'FL), a prevalent HMO and a common infant formula additive, by establishing individualized microbiomes using fecal samples from three infants as the inocula. *Bifidobacterium breve*, a prominent member of infant microbiomes, typically cannot metabolize 2'FL. Using metagenomic data, we predict that extracellular fucosidases encoded by coexisting members such as *Ruminococcus gnavus* initiate 2'FL breakdown, thus critical for *B. breve*'s growth. Using both targeted co-cultures and by supplementation of *R. gnavus* into one microbiome, we show that *R. gnavus* can promote extensive growth of *B. breve* through the release of lactose from 2'FL. Overall, microbiome cultivation combined with genome-resolved metagenomics demonstrates that HMO utilization can vary with an individual's microbiome.

The early-life gut microbiome is a simple yet rapidly changing ecosystem crucial for infant development[1–3]. Early-life events, such as feeding, influence the succession of the gut microbiome[2,4,5]. Breast milk is considered the preferred food source for infants due to its protection against infections and allergy development, among other benefits[6–11]. Many of these advantages can be attributed to human milk oligosaccharides (HMOs), a group of complex carbohydrates unique to human milk that can only be metabolized by certain gut

[1]Department of Plant and Microbial Biology, University of California, Berkeley, CA, USA. [2]Innovative Genomics Institute, University of California, Berkeley, CA, USA. [3]Department of Molecular and Cell Biology, University of California, Berkeley, CA, USA. [4]Department of Chemistry, University of California, Berkeley, CA, USA. [5]Howard Hughes Medical Institute, University of California, Berkeley, CA, USA. [6]Department of Earth and Planetary Science, University of California, Berkeley, CA, USA. [7]Department of Environmental Science, Policy, and Management, University of California, Berkeley, CA, USA. [8]Present address: Department of Environmental Systems Sciences, ETH Zurich, Zurich, Switzerland. [9]Present address: Department of Bacteriology, University of Wisconsin-Madison, Madison, WI, USA. e-mail: jbanfield@berkeley.edu

commensals[12,13]. While breast milk is ideal, it is not accessible to all infants. Within the first 6 months of life, over 70% of infants are estimated to receive formula[14,15], which lacks key breast milk compounds, including HMOs, and can potentially lead to negative health outcomes[16,17].

HMO supplementation is becoming a common practice for shortening the functional gap between infant formula and breast milk[16]. Of the over 200 different HMOs that have been identified in breast milk, a few have been industrially produced and added to infant formula[18]. One of these is the α-1,2-fucosylated trisaccharide 2′-fucosyllactose (2′FL). 2′FL is the most abundant and prevalent HMO in the breast milk of secretor mothers, who consist of ~80% of women globally[19–22]. The presence of 2′FL in feeding has been positively associated with infant health[21]. However, such benefits can be infant-specific, as variations in 2′FL utilization exist between infant gut microbiomes[23–25]. Compositional differences in *Bifidobacterium* spp. partly explain microbiome-specific responses to 2′FL[26] as different *Bifidobacterium* species have varying abilities to metabolize HMOs[27]. Further, variations among strains of the same *Bifidobacterium* species also contribute to interpersonal differences in 2′FL utilization. For instance, only a minority of strains of *Bifidobacterium breve*, an abundant infant gut colonizer, can break down 2′FL by encoding both the α-L-fucosidase and 2′FL transporter[27–30]. Given the natural prevalence and commercial application of 2′FL, we sought to elucidate additional mechanisms that explain microbiome-specific responses to 2′FL.

Prior studies examining 2′FL metabolism mainly focused on *Bifidobacterium*, the primary HMO degrader[27]. 2′FL utilization by coexisting *Bifidobacterium* species suggests the importance of microbial interactions in HMO metabolism[27,31–33]. Besides *Bifidobacterium*, some other community species may also encode genes for 2′FL breakdown and thus could influence *Bifidobacterium*'s growth on 2′FL[34,35]. To date, it remains largely underexplored how co-existing members interact with *Bifidobacterium* in 2′FL assimilation. This is in part due to the lack of tractable gut microbiome models[36]. Experimental microbiome perturbations are usually impossible in humans, and large-scale observational studies are confounded by host-associated factors, such as host genetics and lifestyle, making it difficult to identify causative microbial mechanisms for a given phenotype[37]. Animal models are tractable compared to humans but are expensive, labor-intensive, and have relatively low throughput[38]. Complex continuous culture systems mimicking the human gastrointestinal system have revealed key insights into the gut microbiome ecology but have limited throughput as well[39,40]. It is, therefore, important to establish reasonably representative human microbiome model systems that can be used to explore microbial interactions and their consequences over a wide range of conditions.

Here, to investigate gut consortium interactions contributing to 2′FL metabolism, we established relatively representative in vitro gut microbiomes from biologically unrelated preterm and full-term infants. As we created consortia comprised of natural populations recovered directly from human infant stool samples, our work differs from, but complements, prior in vitro microbiome research and enabled person-specific microbiome analyses[41,42]. Preserving person-specific microbiome signatures, the direct-stool-resuspension approach has previously been applied by others to examine personalized adult gut microbiome drug metabolism[43,44]. In this study, by leveraging functional predictions from metagenomics analyses and testing these predictions experimentally using cultivated microbiomes, we identified crucial microbial functions, which can be carried out by taxonomically distinct organisms in different infants, for influencing the growth of *B. breve* in 2′FL-supplemented media. Our work expands the current understanding of individualized microbiome responses to 2′FL and provides a testing platform for infant-specific nutrient responses, which could, in turn, be leveraged for more targeted and effective infant feeding regimes.

## Results

### Infant gut microbiome cultivation scheme

To elucidate how community interactions may influence *Bifidobacterium*'s growth on HMOs, we selected infant gut microbiomes as inocula based on the presence of a single *Bifidobacterium* species[45]. Microbiomes from three biologically unrelated, breastfed-only infants were chosen, each of which had a distinct composition and complexity, yet they shared a near-identical *Bifidobacterium breve* strain (~99.6% genome-wide average nucleotide identity (gANI)) (Methods). The lack of multiple *Bifidobacterium* species enabled the examination of interactions between non-*Bifidobacterium* and *Bifidobacterium* involved in HMO metabolism.

To explore growth conditions that would allow the maximal recovery of the inoculum species, we began our cultivation efforts with one stool sample ("FT-1") collected from a three-month-old full-term infant. Its microbiome consisted of 8 bacterial species across three phyla (Actinobacteria, Firmicutes, and Proteobacteria) that were ≥0.1% abundant (Supplementary Fig. 1).

### Brain−heart infusion (BHI) with mucin addition recovered representative and stable infant gut microbiomes

To identify a growth medium that allowed the recovery of a microbiome compositionally as close to the original community as possible, we inoculated FT-1 in five complex media (Fig. 1a, b). A compositionally stable community was largely reached after just one passage (Supplementary Fig. 2). Except for human breast milk (HM), a near-complete microbiome was recovered (Fig. 1b and Supplementary Fig. 2). HM, intriguingly, resulted in the least diverse community in which *Bifidobacterium breve* was the only species recovered (Fig. 1b). The lack of detectable organisms in HM-only negative controls indicated *B. breve* was not derived from HM. Additional passaging of the *B. breve*-only HM enrichments in BHI + 0.6% mucin (wt/vol) did not change the community composition (Supplementary Fig. 2b), suggesting *B. breve* was the only viable species in HM enrichments.

Of the non-HM conditions, modified Gifu anaerobic medium (mGAM) had the lowest microbiome recovery rate (Fig. 1b). All species were recovered in at least one replicate of the brain−heart infusion (BHI)-based media, yet the mucin supplementation yielded a community more structurally similar to the inoculum than the BHI-only medium ($P = 0.034$; Wilcoxon rank-sum test on weighted UniFrac Distances) (Fig. 1b). Of the species in the inoculum, *Clostridium neonatale*, present in about a third of the replicates of the BHI-based media with an average abundance of ~1%, had the lowest replicability (Fig. 1b). The stochastic presence of *C. neonatale* in enrichments could be partly due to its low abundance in the inoculum (0.54%; Fig. 1b). In contrast, *Enterococcus gilvus*, undetectable in the initial inoculum, was detected in nearly all non-HM conditions (Fig. 1b). Given its prevalence in enrichments and absence in reagent-only negative controls, we speculated that this species was indeed present in the original stool microbiome but was too low in abundance to be detected (Methods). This led us to conclude the FT-1 microbiome consisted of at least 9 species (Fig. 1b). Overall, while both 0.4% and 0.6% mucin concentrations yielded compositionally stable and reproducible enrichments, 0.6% mucin supplementation resulted in a more stable growth of *B. breve* than 0.4% mucin (Fig. 1c and Supplementary Fig. 2a, b; Supplementary Table 1) (Methods). BHI + 0.6% mucin was thus selected as the base medium for subsequent experiments involving FT-1.

To further test the generality of BHI-mucin media for community cultivation, we inoculated the other two infant fecal samples into BHI with either 0.4% or 0.6% mucin (Methods). One sample ("FT-2"), collected from a 56-day-old full-term infant, consists of at least 20 bacterial species. The other sample ("PT-1"), which was from a 15-day-old preterm infant, has at least six bacterial species in the initial inoculum

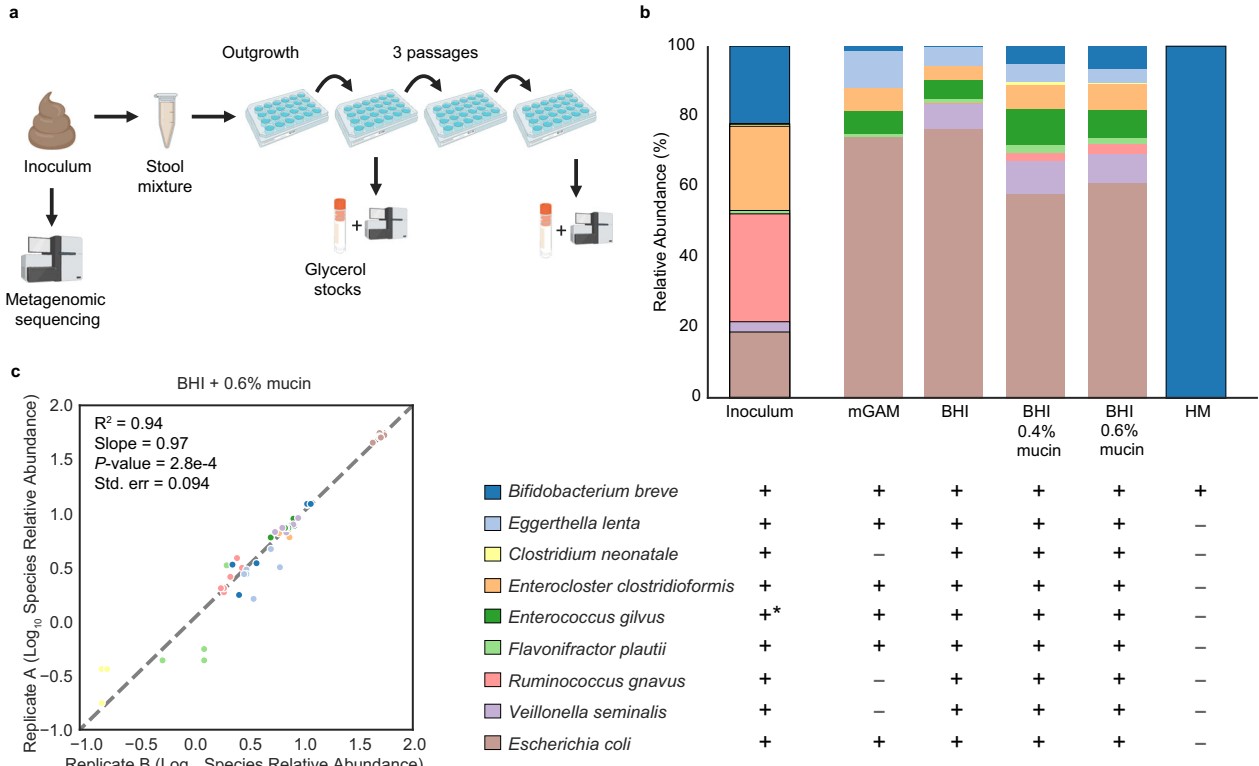

**Fig. 1 | FT-1 microbiome cultivation overview. a** Microbiome cultivation experimental setup. The frozen pilot infant fecal sample was resuspended in a phosphate-buffered saline (PBS) solution before inoculating it into anaerobic media. After an initial 48-hour outgrowth period, the enrichments were subsequently passaged every 48 h for a total of three passages. A fraction of the enrichment was harvested from the 1st and the 3rd passaged enrichments for metagenomic sequencing and long-term storage. Figure created with BioRender. **b** Microbiome compositions grown on different complex media after one passage. Bar height represents normalized species relative abundance, and bars are colored by species. The x-axis represents the growth media, and the data are the averages of all replicates. One biological set (n = 3) was run for modified Gifu anaerobic medium (mGAM),

brain–heart infusion (BHI), and human breast milk (HM). Three and four independent biological replicates (n = 2 or 3 for each independent experiment) were run for BHI + 0.4% mucin and BHI + 0.6% mucin, respectively. "+" and "−" represent the presence and absence, respectively, of the organism detected in the initial FT-1 inoculum. "*" indicates *E. gilvus* was below the detection limit in the initial inoculum and its presence was detected through cultivating in non-HM-rich media. **c** Pairwise species relative abundance comparison among technical replicates. Correlations were calculated using the two-sided linear least-squares regression. Only species in both technical replicates were considered when calculating correlation metrics ($R^2$, slope, P-value, and standard error). Source data are provided as a Source Data file.

(Supplementary Fig. 1). Similar to FT-1, community compositions of FT-2 and PT-1 remained largely unchanged in BHI-mucin over three passages (Supplementary Fig. 3a). Compared to the BHI + 0.6% mucin medium, BHI + 0.4% mucin had a higher microbiome recovery rate, in which 60% and 100% of the FT-2 and PT-1 inocula species, respectively, were recovered, as well as higher replicability (Supplementary Fig. 3b, c). This medium was thus selected for subsequent experiments for these two infant microbiomes.

## HM exerts strong selective pressures on the infant gut microbiome

Despite having distinct initial compositions, all three infant microbiomes consisted of *B. breve* only when grown in HM (Fig. 1b and Supplementary Figs. 2 and 3). The drastic difference in the abundance of *B. breve* grown in HM versus BHI-based media motivated us to test five additional media that contained HM and BHI mixed in different volume proportions (Fig. 2 and Supplementary Fig. 4). Since all three infant gut microbiomes had similar growth outcomes in HM versus non-HM media, FT-1 was selected as the representative for investigating HM's effects on the gut consortia. Cultures were passaged once before composition profiling (Methods).

HM, even a small quantity (i.e., with as little as 10% (vol/vol)), could strongly affect the FT-1 microbiome composition. HM's influence on FT-1 was mostly driven by *B. breve* abundance. As HM concentrations increased, *B. breve* was significantly enriched (Figs. 1, 2 and

Supplementary Fig. 4) (Spearman correlation = 0.86, P = 1.37e−11). Notably, communities became increasingly acidic with rising *B. breve* abundance (Spearman correlation = −0.88, P = 1.29e−12). The changes of pH in *B. breve*-dominated communities (*B. breve* relative abundance >50%) were significantly larger than in communities in which *B. breve* was at low abundance (P = 1.92e−11; Wilcoxon rank sum test) (Fig. 2b).

## 2'FL exerts infant-specific effects on microbiome compositions

Having shown that HM could profoundly shape the infant gut microbiome by promoting the growth of *B. breve*, an effect likely mediated through milk carbohydrates including HMOs, we next investigated if a single HMO, 2'FL, would recapitulate this effect. We included a 2'FL concentration roughly equivalent to that in human milk (0.3% (wt/vol))[13] and a 10x lower one (0.03%) to see if there was a strong concentration-dependent effect on the microbiomes. Communities were grown in BHI-mucin-based media with one passage, as described above (Methods).

When 0.03% 2'FL was added, all enrichments resembled those grown in BHI-mucin-only media (Figs. 1b, 3a–c and Supplementary Fig. 3b, c). When 0.3% 2'FL was supplemented, only the FT-1 community composition changed dramatically (Fig. 3). Comparing 0.3% to the 0.03% 2'FL supplementation, *B. breve* abundance in FT-1, which had a 33% decrease in species richness, increased from ~6% to ~36% (Fig. 3a, d and Supplementary Fig. 5; Supplementary Table 2). Such a community structure remained mostly unchanged when growing the FT-1

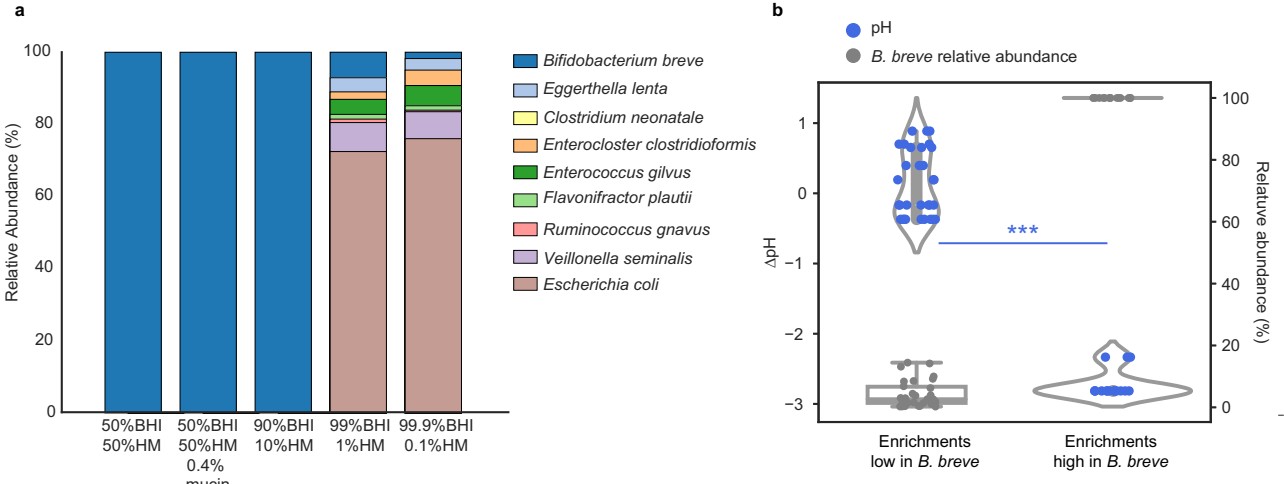

**Fig. 2 | HM poses strong selective pressures on FT-1. a** Community compositions grown on HM-supplemented media with one passage. Bar height represents normalized species relative abundance, and bars are colored by species. The *x*-axis represents the growth media, and the data are averages of the replicates (*n* = 3). **b** Correlation between the change of pH (ΔpH) and *B. breve* abundance for samples grown on HM-supplemented media with one passage. The left *y*-axis represents the change in pH (ΔpH), and the right *y*-axis represents the relative abundance of *B. breve*. Each circle represents a culture replicate and is colored by either ΔpH (blue) or *B. breve* abundance in that culture (gray). The *x*-axis represents the community type, either dominated (*n* = 16 technical replicates from 6 independent experiments) or not dominated (*n* = 30 technical replicates from 11 independent experiments) by *B. breve* (\*\*\**P* = 1.92e−11; Wilcoxon rank sum test). The box plot shows the interquartile range (IQR) of *B. breve* relative abundances in different growth media with the central line representing the median; the whiskers extend from the lower and upper quartiles to 1.5 times the IQR. Similarly, the violin plot shows the distribution of ΔpH in cultures enriched or depleted with *B. breve*; the inner box shows the IQR. Source data are provided as a Source Data file.

community in media supplemented with 2′FL concentrations higher than 0.3% (Fig. 3f and Supplementary Fig. 5). Notably, when removing glucose, whose concentration is 0.3% (wt/vol) in the BHI used for community cultivation, the abundance of *B. breve* increased further (*q* = 0.025; two-sided Welch's *t*-test adjusted for multiple comparisons using Benjamini–Hochberg correction), reaching ~72% (Supplementary Fig. 6). The community diversity of FT-2 remained the same regardless of 2′FL concentrations, although the abundances of several species, including *B. breve*, changed significantly (Fig. 3b, e; Supplementary Table 2). The community composition and diversity of PT-1 stayed the same across different 2′FL concentrations (Fig. 3c).

The changes in community compositions in response to 2′FL concentrations were also reflected in the pH. In media that contained 0.03% 2′FL, the pH values of all enrichments were ~7.1. However, with a 2′FL concentration of 0.3%, the pH of the FT-1 and FT-2 enrichments dropped by 2.1 and 1.0, creating an acidic milieu, whereas the pH of PT-1 remained the same (Fig. 3a–c). Based on these findings, we infer that the effects of 2′FL at the tested concentration levels can indeed be infant-specific, supporting previous observational studies[23–25]. In certain cases, it can lead to a significant enrichment of gut commensals, such as *B. breve*.

**In FT-1, *B. breve* could utilize 2′FL in the presence of *R. gnavus***

To elucidate why 2′FL supplementation resulted in distinct effects on these infant microbiomes, we assessed the genomically defined metabolic potentials of the communities, focusing primarily on FT-1 because it exhibited the most substantial change at the physiologically relevant 2′FL concentration of 0.3%. All genomes in FT-1 were reconstructed, and functional predictions were made using CAZyme, KEGG, Pfam, TCDB, and signalP (Methods).

We first examined whether *B. breve* in FT-1 could directly metabolize 2′FL by isolating it from FT-1 and growing it in media with 2′FL as the sole carbohydrate source (Methods). No growth was detected (Supplementary Fig. 7). Genomic analysis revealed that *B. breve* encoded one fucosidase, GH95, but this gene was predicted to be intracellular due to the lack of a signal peptide. With the lack of 2′FL

transporters, we concluded that *B. breve* from FT-1 cannot metabolize 2′FL (Methods).

Since *B. breve* cannot break down 2′FL, we hypothesized that the *B. breve*-promoting effect of 2′FL seen in FT-1 resulted from other community members liberating lactose and fucose from 2′FL, which could subsequently be metabolized by *B. breve*. A community-wide CAZyme survey revealed that *R. gnavus* was the only organism besides *B. breve* in FT-1 whose genome encodes fucosidases (Methods). Specifically, it encodes six fucosidase genes (three GH29 and three GH95 genes), two of which (one GH29 and one GH95 gene) were predicted to be extracellular. To investigate whether the presence of *R. gnavus* was sufficient for *B. breve* to grow on 2′FL, we isolated *R. gnavus* from FT-1 and inoculated this strain as well as the *B. breve* strain isolated from FT-1 as mono- and co-cultures in media with 2′FL or its constituents, L-fucose or D-lactose, as the sole carbohydrate source (Methods). The growth was measured using optical density (OD, *λ* = 600 nm), and the coculture compositions were assessed via amplicon sequencing (Methods).

The *R. gnavus* monoculture grew on 2′FL, but not the *B. breve* culture (Fig. 4a). The co-culture had a higher growth rate and yielded a higher final cell density than the *R. gnavus* monoculture (Fig. 4a; Supplementary Table 3). Notably, *B. breve* dominated the co-culture community (~80% of the community) (Fig. 4a stacked bar chart), paralleling the findings of the FT-1 community grown on 2′FL (Fig. 3a).

*R. gnavus* but not *B. breve* could grow on L-fucose (Fig. 4b). The growth rate and the final cell density of the co-culture were similar to those of the *R. gnavus* monoculture grown on fucose (Fig. 4b; Supplementary Table 3). In contrast to growth on 2′FL, this co-culture was dominated by *R. gnavus* (Fig. 4b stacked bar chart).

Both the *R. gnavus* and *B. breve* monocultures grew on D-lactose (Fig. 4c). *B. breve* had a higher growth rate and reached a higher final cell density than *R. gnavus* (Fig. 4c; Supplementary Table 3). The co-culture's growth rate and the final cell density were comparable to those seen in the *B. breve* monoculture (Fig. 4c; Supplementary Table 3), and the community was dominated by *B. breve* (Fig. 4c stacked bar chart).

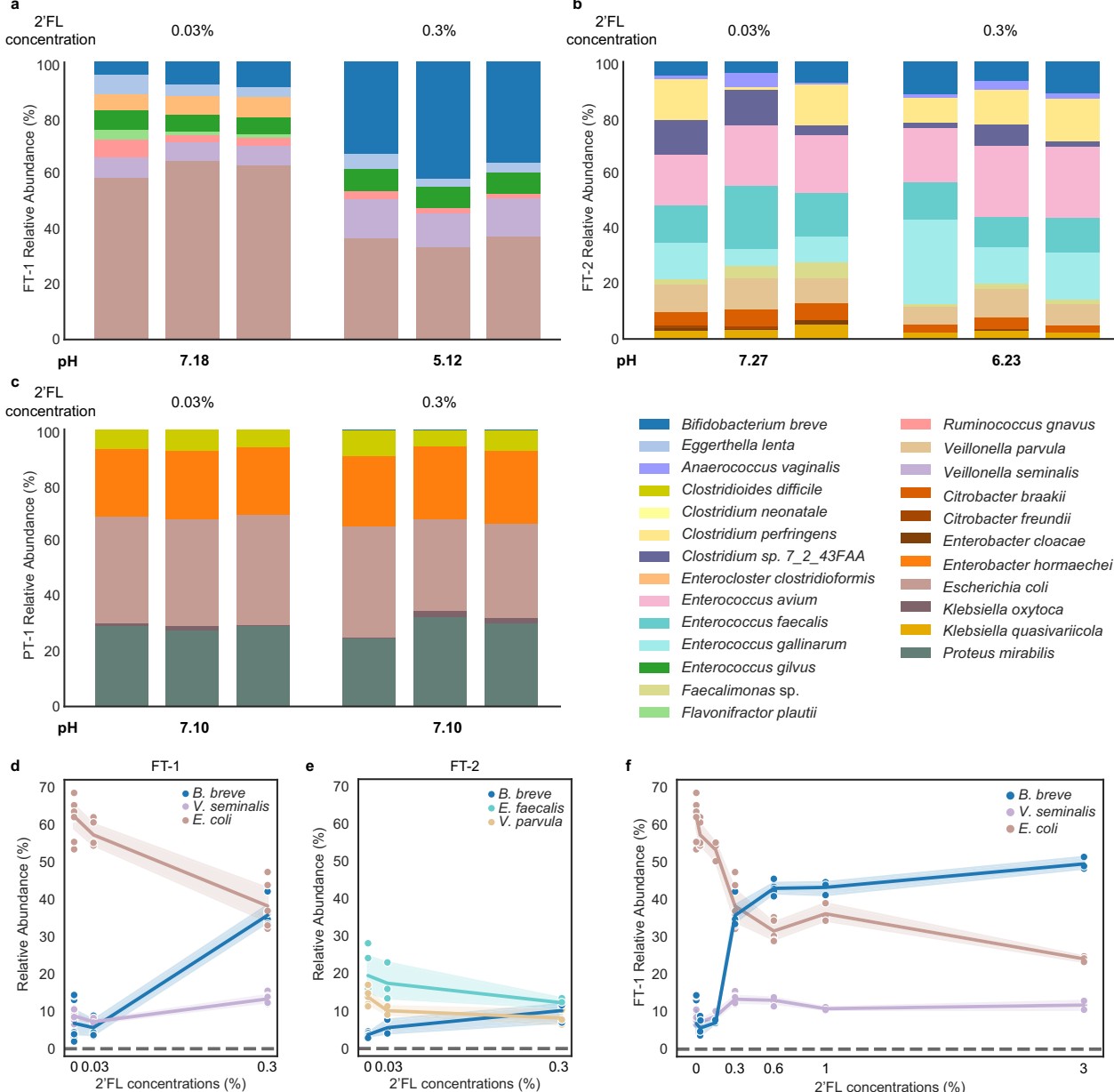

**Fig. 3 | 2'FL supplementation resulted in distinct responses from different infant gut microbiomes. a–c** The compositions of the FT-1 (**a**), FT-2 (**b**), and PT-1 (**c**) enrichments grown in BHI-mucin supplemented with 0.03% and 0.3% 2'FL with one passage. Bar height represents normalized species relative abundance, and bars are colored by species. The *x*-axis represents the growth media, and all replicates are shown (*n* = 3). The pH of each enrichment is the mean across the three replicates. The figure legend is shown on the bottom right of the figure panel. **d, e** The significant enrichment or depletion of FT-1 (**d**) and FT-2 (**e**) species cultured in different 2'FL concentrations (0, 0.03%, and 0.3%) with one passage. Each circle represents a species whose abundance changed significantly in different 2'FL concentrations (Spearman correlation |*r*| ≥ 0.8 and *q* < 0.05; Supplementary Table 2), and the line represents the average of all replicates; error bands indicate the 95% confidence interval. The *x*-axis represents the 2'FL concentrations

(replicate numbers for FT-1: *n* = 9 from 4 independent experiments for 0% 2FL, *n* = 5 from 2 independent experiments for 0.03% 2'FL, *n* = 6 from 3 independent experiments for 0.3% 2'FL; for FT-2: *n* = 3 from 1 independent experiment for all 2'FL concentrations). **f** The FT-1 inoculum grown in media with up to 3% 2'FL with one passage and species that showed a significant depletion or enrichment in (**d**) are shown. The line represents the average of all replicates; error bands indicate the 95% confidence interval. The *x*-axis represents 2'FL concentrations (replicate numbers: *n* = 9 from 4 independent experiments for 0% 2FL; *n* = 5 from 2 independent experiments for 0.03% 2'FL; *n* = 3 from 1 independent experiment for 0.15% 2'FL; *n* = 6 from 3 independent experiments for 0.3% 2'FL; *n* = 5 from 2 independent experiments for 0.6% 2'FL; *n* = 3 from 1 independent experiment for 1% 2'FL; *n* = 3 with 1 independent experiment 3% 2'FL). Source data are provided as a Source Data file.

Overall, the experiments revealed that *R. gnavus* grows on 2'FL, lactose, and fucose, whereas *B. breve* can only grow on lactose. Given that *B. breve* did not show extensive growth on 2'FL or fucose, we infer that *B. breve*'s growth in the co-culture grown on 2'FL was due to the lactose released by *R. gnavus*. As the only member that can break down 2'FL, *R. gnavus* is likely an essential FT-1 member when grown on 2'FL. The community grown on ≥0.3% 2'FL was dominated by *B. breve* and *E.*

*coli*, followed by *V. seminalis*, *E. gilvus*, *E. lenta*, and last, *R. gnavus* (Fig. 3a). We thus infer that *R. gnavus* is a keystone species, and it modulates the microbiome by initiating the breakdown of 2'FL otherwise inaccessible to the remaining community members. *E. coli* has the potential to metabolize lactose and fucose, *E. gilvus* is predicted to metabolize lactose, and *V. seminalis* likely metabolizes fucose. *E. lenta* did not encode genes for metabolizing either sugar,

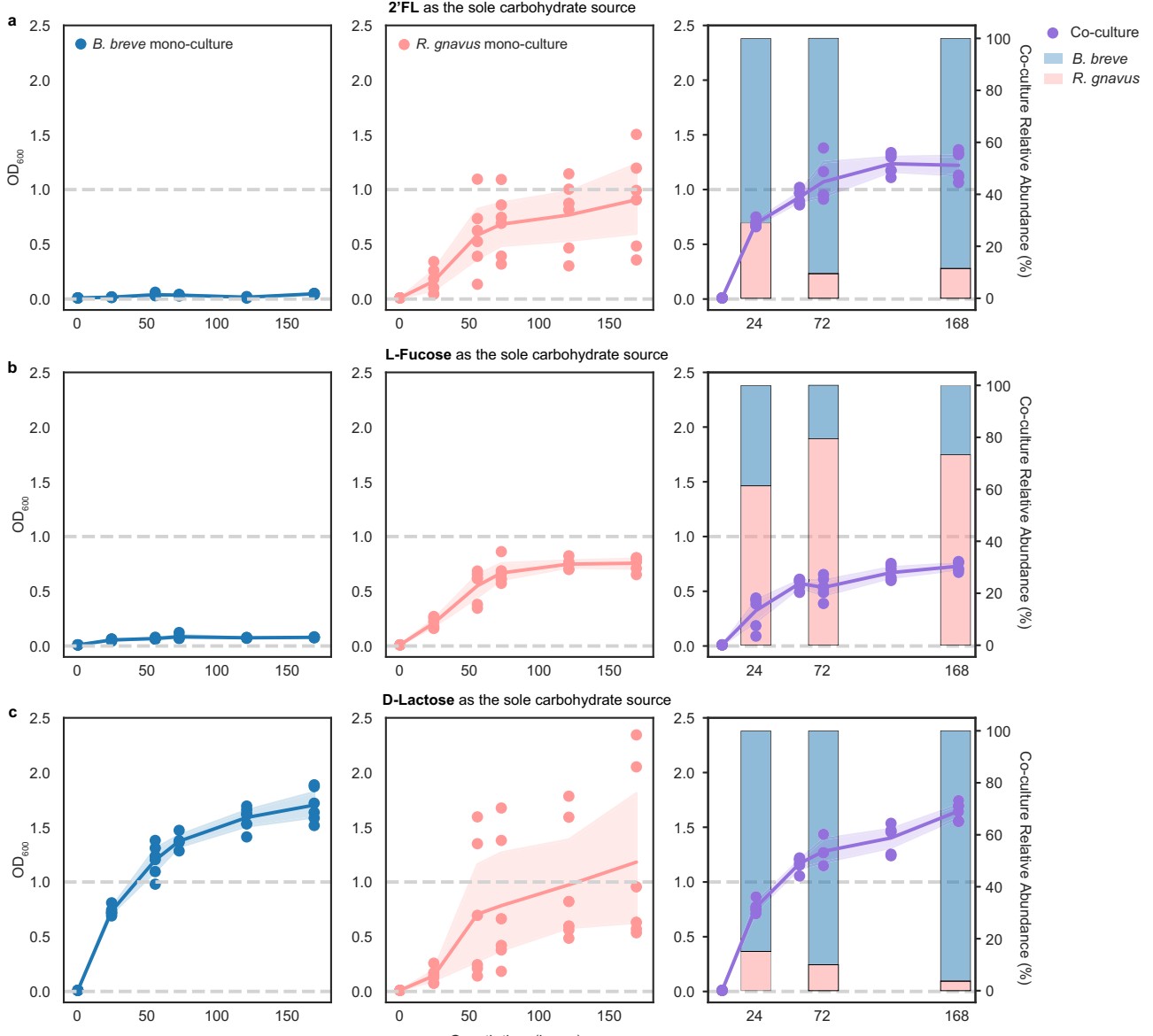

**Fig. 4 | Mono- and co-culture growth on 2'FL, L-fucose, and D-lactose.** *B. breve* (blue) and *R. gnavus* (pink) monocultures and cocultures (purple) grew in media in which 2'FL (**a**), L-fucose (**b**), or D-lactose (**c**) was the only carbohydrate source. The *x*-axis represents the growth time (in hours). The left *y*-axis represents OD$_{600}$, and the right *y*-axis in the co-culture panels represents the normalized relative abundance, calculated using sequencing coverage and shown as the averages of all replicates ($n = 6$). The experiment was done in triplicates and was repeated once. Each circle represents a mono- or co-culture replicate sample and the line represents the average of all replicates at each time point; error bands indicate the 95% confidence interval. The relative abundance of each species (*B. breve* or *R. gnavus*) at the 24th, 72nd, and 168th hours was shown in stacked bar charts in less saturated colors in the co-culture panels. Source data are provided as a Source Data file.

suggesting it potentially utilizes by-products from the FT-1 community (Methods) (Supplementary Data File 1).

### Supplementation of *R. gnavus* and 2'FL shifts a preterm microbiome into a full-term-like, *Bifidobacterium*-enriched community

Given the dependency of *B. breve* on *R. gnavus* in FT-1 grown on 2'FL, we sought explanations for the extensive growth of *B. breve* in FT-2 and the limited growth in PT-1. The presence of the intracellular GH95 fucosidase and the lack of 2'FL transporters suggest none of the *B. breve* strains in these inocula could directly utilize 2'FL. Thus, we applied the same genome-resolved method used for FT-1 to the FT-2 and PT-1 communities (Methods).

Three species (*Clostridium perfringens*, *Clostridium* sp. 7_2_43FAA, and *Faecalimonas* sp.) in FT-2 are predicted to metabolize 2'FL

extracellularly (Methods) (Supplementary Data File 2). Since the growth of *B. breve* on 2'FL was less significant in FT-2 than in FT-1, we speculated that other factors, such as inter-species competition, might result in limited *B. breve* growth. The extensive growth of *B. breve* in FT-1-based mono- and co-cultures, when grown in media with lactose as the sole carbohydrate (Fig. 4a, c), led us to hypothesize that other organisms in FT-2 might be directly competing against *B. breve* for lactose. Indeed, we found that the genomes of the three extracellular fucosidase-encoding species also encode extracellular β-galactosidases (Supplementary Data File 2), some of which are genomically adjacent to the extracellular fucosidases, suggesting these organisms could further break down extracellular lactose freed from 2'FL into glucose and galactose. No extracellular β-galactosidases were detected in the FT-1 or the PT-1 community. We thus infer that the lower abundance of *B. breve* in FT-2 compared to FT-1 in

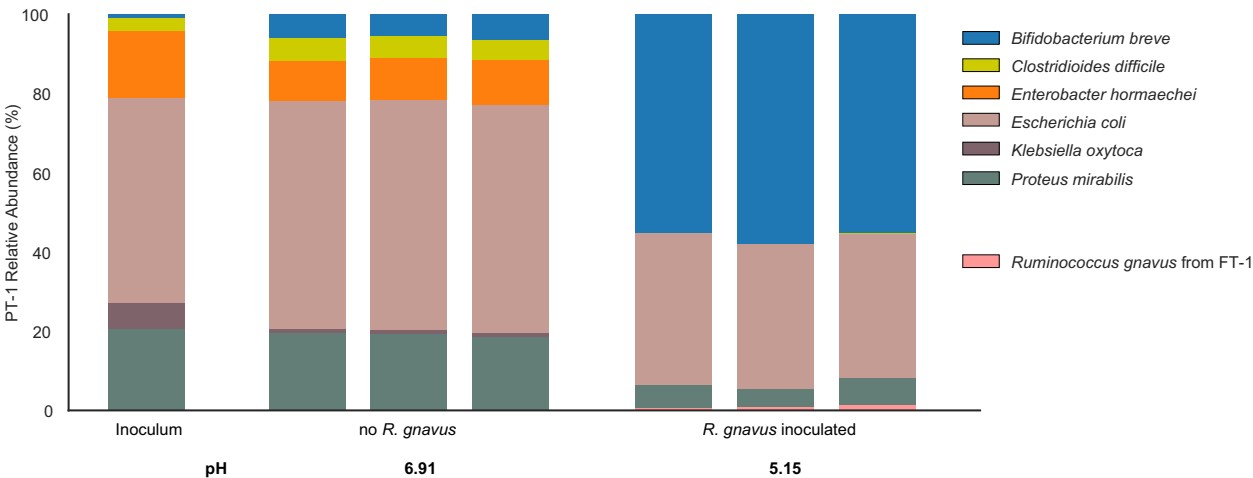

**Fig. 5 | Adding *R. gnavus* to PT-1 enriched *B. breve* in 2′FL-supplemented media.** The PT-1 community with or without the supplementation of *R. gnavus* was grown in media containing 0.3% 2′FL with one passage. Bar height represents normalized species relative abundance, and bars are colored by species. The x-axis represents the community type, and all replicates are shown. Source data are provided as a Source Data file.

2′FL-supplemented media may be the consequence of the degradation of lactose, *Bifidobacterium*'s preferred carbohydrate source[46,47], by other community members, which reduces the amount of lactose that can be used by *B. breve* for growth.

In PT-1, no other organisms besides *B. breve* encode a fucosidase. This explains why the PT-1 community failed to respond to 2′FL supplementation and remained dominated by Proteobacteria (Fig. 3c). To test whether adding a strain encoding extracellular fucosidases would be sufficient to enrich *B. breve* in PT-1 when sufficient 2′FL is supplemented, we inoculated *R. gnavus* isolated from FT-1 to PT-1 (Methods). Notably, *B. breve* was significantly enriched in the *R. gnavus*-positive PT-1 microbiome grown in media with 0.3% 2′FL and reached an abundance level similar to that seen in FT-1 (Figs. 3a and 5). As in the FT-1 enrichment, the increase in *B. breve* abundance in the *R. gnavus*-supplemented PT-1 microbiome was correlated with a decrease in community species diversity and pH, reaching an average value of 5.15, ~1.8 points lower than the unsupplemented PT-1 microbiome grown under the same conditions (Fig. 5).

## Discussion

Diet affects gut microbiome compositions and functions[1,48]. As each human is colonized by a distinct gut microbiome, it is important to study dietary effects on different microbiomes to uncover the diversity of relevant interactions and commonalities across individuals[49,50]. Here, using clinically relevant stool samples of human infants, we established stable, reproducible, and individualized laboratory gut consortia and investigated inter-species interactions contributing to microbiome-specific responses to one of the most abundant human milk oligosaccharides (HMOs)[51], 2′-fucosyllactose (2′FL). We found that, despite being unable to break down 2′FL on its own, *B. breve* could grow well so long as it co-exists with species that encode extracellular fucosidases. The influence of extracellular versus intracellular metabolism of HMOs on microbiomes has also been seen in other infants, as well as in adults[26,31,52,53]. This led us to propose that glycosidase repertoires, as well as their cellular localization, should be investigated when assessing complex carbohydrate metabolism. Overall, leveraging in vitro cultivation and genomic analyses using distinct infant microbiomes, we showed that 2′FL, if supplemented into functionally matching microbiomes, could exert similar bifidogenic effects as HM in shaping gut microbiomes (Fig. 6).

To our knowledge, this is the first study in which human milk (HM) is directly used to cultivate infant gut microbiomes. *B. breve*, regardless

of the initial abundance and community composition, reached nearly 100% abundance in all HM enrichments. By experimentally showing a *B. breve*-dominant community can be established using media containing as little as 10% HM, we demonstrated the significant role HM plays in shaping gut microbiome compositions. The enrichment of *B. breve* in HM-based media aligns with findings from observational studies that breastfeeding results in a *Bifidobacterium*-enriched gut microbiome with a lower species diversity when compared to that of formula-fed infants[4,54–56]. The bifidogenic effect of HM is partly due to *Bifidobacterium* efficiently fermenting breast milk carbohydrates, creating a low-pH environment unsuitable for the growth of many members, including potential pathogens (i.e., Proteobacteria)[57,58]. Indeed, consistent with prior in vivo studies[28,33,59–63], in our enrichments, we also observed a negative correlation between *B. breve* abundance and the pH, as well as species richness. The near-complete dominance of *B. breve* in HM enrichments slightly deviates from the natural gut microbiomes of breastfed infants. We speculate the greater prevalence of *B. breve* in HM enrichments compared to the infant's gut is partly due to the higher concentration of lactose, which is mostly absorbed in the infant's small intestine[64]. Using *B. breve* isolated from an infant gut consortium, we showed that this species could efficiently use lactose and grow on it as the sole carbohydrate source. Further supporting our hypothesis is the observation that extracellular β-galactosidases-encoding species, which degrade lactose, could limit the growth of *B. breve*. Overall, our cultivated microbiomes largely recapitulated the bifidogenic in vivo responses to breastfeeding.

Stable colonization of *Bifidobacterium* is crucial in early-life gut microbiome assembly and immune system development[65,66]. Compared to full-term, preterm infant gut microbiomes are depleted of persisting *Bifidobacterium* strains[45,67]. Besides clinical factors such as frequent antibiotic exposures and lack of sufficient breastfeeding[68,69], microbiome compositions and functions, such as those dominated by Proteobacteria, could also negatively affect the colonization of *Bifidobacterium*, especially those with a limited HMO metabolic capacity, including *B. breve*. Supplementation of *Bifidobacterium* probiotic strains with a broad HMO metabolic capacity, such as *B. longum* subsp. *infantis*, thus has been used to rescue dysbiotic preterm gut microbiomes[57,59,60,65,70]. However, the efficacy of exogenously administered *Bifidobacterium* strains can be affected by the resident microbiome[3,30,71]. For instance, despite having a limited HMO-metabolism ability, *B. breve* can outcompete other *Bifidobacterium* species, including *B. infantis*, if it colonizes the infant gut first[71]. By enriching indigenous *B. breve* using a targeted combination of 2′FL and

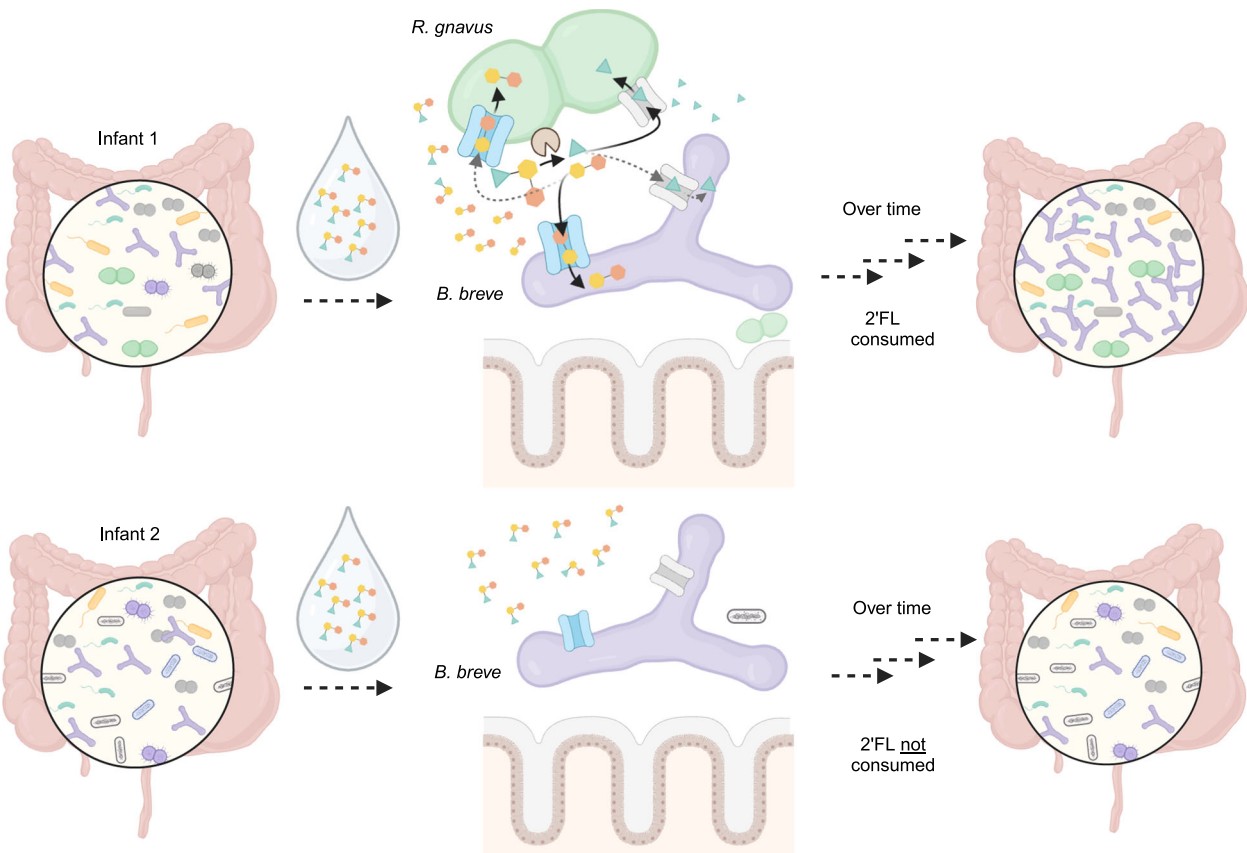

**Fig. 6 | 2′FL exerts beneficial effects on functionally matching microbiomes.** Infant 1 and Infant 2 both consisted of a *Bifidobacterium breve* strain (purple) that cannot metalize 2′FL itself. The main difference between these two infant microbiomes is the presence (Infant 1) or absence (Infant 2) of an extracellular-fucosidase-encoding *Ruminococcus gnavus* strain (green). When feeding these two infants with the same 2′FL-supplemented drink, only Infant 1's gut microbiome was positively modified. This is mainly due to *R. gnavus* breaking 2′FL into fucose and lactose, which can be used by *B. breve* and other community members for growth. Infant 2, which lacks *R. gnavus*, minimally changed following feeding. Figure created with BioRender.

*Ruminococcus gnavus*, our study provided an alternative to probiotic *Bifidobacterium* administration.

*R. gnavus* is prevalent in both healthy infant and adult gut microbiomes and their capability in metabolizing complex carbohydrates, including mucin and HMOs, partly contributes to their stable gut colonization since early-life[72]. In our study, microbiome cultivation using 2′FL revealed *R. gnavus* as a keystone species. Keystone species have a disproportionately large importance in ecosystems relative to their abundance[73,74]. *R. gnavus*'s disproportionately large impact on some microbial communities thus makes it a potential ideal candidate for microbiome intervention. However, additional work is needed for identifying specific *R. gnavus* probiotic strains as some members of this species were found to be strongly associated with diseases including inflammatory bowel diseases (IBD)[72,75].

Overall, our work demonstrates the potential importance of personalized microbiome interventions. Specific probiotics and/or prebiotics may reshape the gut microbiomes of distinct infants differently, thus influencing the subsequent gut microbiome assembly as well as infant development[76]. Widely prescribed probiotics show some positive outcomes, including reducing the chance of developing necrotizing enterocolitis (NEC) and sepsis in preterm infants[77,78], yet their efficacies often vary[25,76]. For instance, *B. breve* is a commonly used probiotic[30,79]. However, partly due to its inability to metabolize many HMOs, including fucosylated saccharides such as 2′FL, varied outcomes in infants have been reported[80–82]. Such variability among individuals also exists when using multi-strain probiotic cocktails[83,84]. For instance, in a recent clinical trial, less than 40% of the preterm infant gut microbiomes responded positively to a five-strain probiotic treatment[84]. We experimentally showed that the resident gut microbiome represents a key factor in determining the treatment outcome. As shown with the preterm gut microbiome here, only when consortium-matching prebiotics and/or probiotics are administered, the expected health-promoting outcomes can be achieved. In addition to the gut microbiome, other clinical factors such as maternal secretor status, diets and the immune system could affect the efficacy of microbiome interventions74. Compared to secretors, non-secretor mothers lack the allele to produce 2′FL and other α1-2 fucosylated oligosaccharides[85]. In our study, maternal and infant secretor statuses were not collected. In future studies, it would be interesting to investigate how maternal and infant secretor statuses influence the outcome of microbiome interventions.

In summary, by cultivating different infant gut microbiomes, we showed that community compositions influence HMO metabolism. The microbiome cultivation systems established here can be extended to experimentally elucidate individualized responses to other perturbations, such as antibiotics and other complex carbohydrates. The model microbiomes developed in this work can also be used in microbiome editing-based experiments to study mechanisms of inter-organism interactions in communities. For instance, we recently developed a CRISPR–Cas-based approach to edit organisms in the context of their microbiomes and demonstrated strain-specific editing sensitivity in an in vitro infant gut model[86].

The gut microbiome interacts with supplemented compounds bidirectionally, in so far as the compounds impact the microbiome composition and the microbiome members impact the compounds[87]. Significant impacts of these interactions on our health can arise

through microbiome processing of dietary compounds, alteration of the efficacy of pre- and probiotics, and moderation of the impact of prescription medications[87]. As each person is colonized by a distinct gut microbiome, it may be possible to maximize the effectiveness of an intervention or treatment by evaluating a compound's impact on the individual's microbiome in a laboratory microbiome experiment. Overall, our findings demonstrate a way in which gut microbiome experiments can be an integral part of precision medicine. Insights gained from these individualized in vitro microbiomes could enable more precise and effective treatments.

## Methods

### Study protocol
This study was reviewed and approved by the University of Pittsburgh Human Research Protection Office (IRB STUDY19120040). All infant stool samples were collected with parental consent and subjects were de-identified before the receipt of samples.

### Anaerobic growth media selection and preparation
BHI was selected as the primary base since this medium allowed the isolation and stable maintenance of adult gut microbiomes in previous studies[42,88]. We also included the modified Gifu anaerobic medium (mGAM), another rich medium used to isolate individual gut microbes and consortia[43,89]. Given that all three infants were only fed human breast milk (HM), we also included HM as a growth medium. Mucin was supplemented since it can be a key carbon source for some gut species such as *R. gnavus*[90]. For cultivating *Bifidobacterium breve* isolates, MRS broth was used.

Aerobic BHI (BBL Supplier No. 211059), BHI without dextrose (Bioworld Supplier No. 306200141), mGAM (HiMedia SKU M2079-500G), and MRS (Sigma-Aldrich SKU 69966-500G) were prepared using Milli-Q water as instructed by the suppliers. For preparing 0.4% or 0.6% (wt/vol) mucin-supplemented BHI media, we added 4 or 6 g of mucin (type III mucin from the porcine stomach; Sigma-Aldrich SKU M1778-100G) to 37 g of BHI powder before mixing with 1000 mL Milli-Q water. Aerobic HM (pooled from multiple donors; Lee Biosolutions, Inc) was prepared by centrifuging at 5000×*g* for 5 min at 4 °C. After centrifuging, the supernatant was saved and all solids (cream and pellets) were discarded..

Subsequently, all aerobic media solutions were sparged with $N_2$ to remove oxygen and supplemented with L-cysteine hydrochloride (0.5 g/L) (Sigma-Aldrich SKU C1276-10G) and 0.1% (wt/vol) resazurin (1 mL/L) (Sigma-Aldrich SKU R7017-1G) as a reductant and oxygen indicator, respectively, prior to autoclaving.

To prepare carbohydrate stock solutions (2'FL, L-fucose, and D-lactose), we dissolved 10 g of the respective carbohydrate in 100 mL Milli-Q water and vacuum filtered (Corning™ 431475). Sugar solutions were loosely capped and stored in the anaerobic chamber for at least 48 h before mixing with BHI-based media for cultivation. 2'FL (CARE4U 2'FL) was gifted by DuPont Nutrition & Biosciences, and L-fucose and D-lactose were purchased from Fisher Scientific (Catalog No. AAA1678906) and Sigma-Aldrich (SKU 17814-1KG), respectively.

### Infant gut microbiome selection and cultivation
FT-1 and FT-2 were collected from males and PT-1 was collected from a female. This information was reported by parents during stool sample collection. Infant's sex and gender, race, ethnicity, or other socially relevant grouping were not considered when selecting samples for cultivation. Stool inocula were chosen solely based on microbiome compositions, which were all collected and analyzed in our previous study[45]. Detailed information on infant metadata and infant enrollment/sample collection can be found in Lou et al.[45].

All cultivation work was performed in the same anaerobic chamber (Coy Labs; 70% $N_2$, 25% $CO_2$, 5% $H_2$). The infant stool inoculum, stored in the −80 °C freezer, was resuspended in phosphate-buffered

saline (PBS; Thermo Fisher Scientific Catalog Number 10010023) in a 1:2 wt/vol ratio. The resulting stool resuspension was homogenized by pipetting, and 15 μL of this mixture was added into 3 mL of a growth medium in a 24-deep-well block. The culture was allowed to recover for 48 h at 37 °C with shaking, after which it underwent three more passages of 30 μL into 3 mL of fresh liquid medium, with each allowed to grow for 48 h at 37 °C. A cell-free negative control was included for each growth medium. After the first and the third (final) passaging, a 500-μL aliquot of the culture was taken to make 25% (vol/vol) glycerol stock and a 1.5-mL aliquot was pelleted for DNA extraction and metagenomic sequencing. The remaining culture was used for pH measurements using a handheld pH meter (Apera model PH60S with the Swiss Spear pH Electrode (±0.01 pH accuracy)).

### DNA extraction, library preparation, and metagenomic sequencing for cultivated microbiomes
DNA extraction of the frozen cultivated microbiomes and/or fecal samples was performed using the Qiagen DNeasy PowerSoil DNA Isolation Kit (Catalog No. 47016) as instructed by the manufacturer. Library preparation was performed using the NEBNext Ultra II FS DNA Library Prep Kit (Catalog No. E7805L) as instructed by the manufacturer. Final sequencing-ready libraries were visualized using the Agilent 2100 Bioanalyzer before pooling, and the final pooled library was quantified through qPCR using the KAPA Library Quantification Kit (Catalog No. 7960336001) as instructed by the manufacturer. Sequencing was performed using Illumina NextSeq 1000/2000 150 paired-end sequencing lanes with 10% PhiX spike-in controls. Post-sequencing bcl files were converted to demultiplexed fastq files per the original sample count with Illumina's bcl2fastq v2.20 software.

### Mono- and co-culture experiments
All work was performed in the same anaerobic chamber (Coy Labs; 70% $N_2$, 25% $CO_2$, 5% $H_2$) as the microbiome cultivation experiments described above.

*Bifidobacterium breve* and *R. gnavus* were isolated from FT-1 using Bifidobacterium-Selective Agar (BSA; Anaerobe Systems AS-6423). Specifically, the BHI + 0.6% mucin and HM enrichments were streaked from frozen glycerol stocks onto four BSA plates (two per enrichment type) and incubated at 37 °C for 48 h. Forty colonies (10 per plate) were selected, streaked, and incubated at 37 °C for 48 h on BSA plates two additional times. The full-length 16 S rRNA gene was amplified via colony PCR using primers 27 F (5′-AGRGTTYGATYMTGGCTCAG-3′) and 1391 R (5′-GACGGGCGGTGWGTRCA-3′) (Taq DNA Polymerase with Standard Taq Buffer kit (New England Biolabs M0273S)). The thermocycling conditions used were as follows: 1 cycle of 95 °C for 3 min, 30 cycles of 95 °C for 30 sec, 55 °C for 30 sec, and 72 °C for 45 sec, and 1 cycle of 72 °C for 10 min. The amplicon was sequenced using Sanger sequencing, and colony taxonomy was identified using BLASTN against the 16 S rRNA sequences from the NCBI BLAST Databases (downloaded in September 2022). Five colonies from each strain (*B. breve* and *R. gnavus*) were grown in MRS and BHI + 0.6% mucin media, respectively, for 48 and 24 h, respectively, before suspending in 50% (vol/vol) glycerol for making 25% glycerol stocks and were stored at −80 °C. For all subsequent experiments, to ensure the consistency of the data, we used only one colony from each strain. The remaining four from each strain were stored as backups.

Prior to the mono- and co-culture experiments, *B. breve* and *R. gnavus* were inoculated from glycerol stocks and were grown anaerobically in 6 mL MRS and BHI + 0.6% mucin media, respectively, for 48 and 24 h, respectively. The cells were washed twice using anaerobic PBS by centrifuging at 2000 x g for 10 min at room temperature, the supernatants were discarded, and the cells were resuspended in 1.5 mL of PBS. $OD_{600}$ was measured for the resuspended cultures. For the monoculture experiments, we inoculated an $OD_{600}$ of 0.1 of the culture into the corresponding growth media (BHI without dextrose

solution supplemented with 1% (wt/vol) of 2'FL, L-fucose, or D-lactose), and for the co-culture experiments, an equal $OD_{600}$ of 0.05 of resuspended *B. breve* and *R. gnavus* cultures were inoculated into the same set of growth media as the monoculture experiments. All cultures were grown in triplicates for 144 h, and $OD_{600}$ was measured every 24 and/or 48 h using Tecan microplate readers. Mono- and co-cultures were harvested 24, 72, and 144 h after inoculation for community composition assessment. The mono- and co-culture growth rates were calculated as the slope of the natural log of the $OD_{600}$ in the exponential phase over time.

## DNA extraction and amplicon sequencing for the mono- and co-culture experiments

DNA extraction of the mono- and co-cultures was performed via the Qiagen DNeasy UltraClean 96 Microbial Kit (Catalog No. 10196-4). Relative abundance was estimated by amplicon sequencing and corrected by the 16 S rRNA gene copy number of each species (type strains of *B. breve* and *R. gnavus* encode 2 and 5 copies of 16 S rRNA genes, respectively)[91]. The V1–V2 regions of the 16 S rRNA gene were amplified using a forward (5'-TGCTTAACACATGCAAGTCG-3') and a reverse primer (5'-TCTCAGTCCCAATGTGGCCG-3'). The Phusion® High-Fidelity PCR Master Mix (New England Biolabs Catalog No. M0531S) was used for PCR amplification. The thermocycling conditions used were as follows: 1 cycle of 98 °C for 30 sec, 35 cycles of 98 °C for 10 sec and 72 °C for 10 sec, and 1 cycle of 72 °C for 5 min. Amplicons were cleaned using 1.8X SPRIselect magnetic beads (Beckman Coulter Product No. B23318).

## *R. gnavus* microbiome supplementation assay

*R. gnavus* isolated from FT-1 and PT-1 enrichment passaged in BHI + 0.4% mucin + 0.3% 2'FL were inoculated from the glycerol stocks and were grown anaerobically in 6 mL BHI + 0.6% mucin and BHI + 0.4% mucin + 0.3% 2'FL media, respectively, for 24 h and 48 h, respectively. The *R. gnavus* and the PT-1 enrichment glycerol outgrowths were then mixed in a 1:4 vol/vol ratio, and 30 uL of the resulting mixture was added into 3 mL of a growth medium in a 24-deep-well block. The remaining procedure followed the standard community cultivation assay (see the Method section "Infant gut microbiome cultivation").

## Read-mapping-based species detection

Reads from all enrichments were trimmed using Sickle (www.github.com/najoshi/sickle), and those mapped to the human genome with Bowtie2[92] (v2.3.5.1) under default settings were discarded. Subsequently, reads from each enrichment were mapped to the 1005 representative subspecies (generated from Lou et al.[45]). inStrain (v1.5.1) *profile*[93] was run on all resulting mapping files using a minimum mapQ score of 0. Genomes with ≥0.5 breadth (meaning at least half of the nucleotides of the genome are covered by ≥1 read) with a minimum 0.1% relative abundance (number of reads mapped to the given genome divided by the total number of paired-end reads) in samples were considered present. If organisms were not detected in the initial inocula and/or negative controls and were present in <40% of the replicates across all tested growth conditions, they were defined as contaminants and were removed from the final data tables.

## Metagenomic assembly and gene prediction

Reads from each sample were assembled independently using IDBA-UD[94] (v1.1.3) under default settings. Co-assemblies were also performed for each infant sample, in which reads from all enrichments of that infant sample were combined and assembled. Scaffolds that are <1 kb in length were discarded. The remaining scaffolds were annotated using Prodigal[95] (v2.6.3) to predict open reading frames using default metagenomic settings.

## Metagenomic de novo binning

Pairwise cross-mapping was performed between all enrichments from each infant sample to generate differential abundance signals for binning. Each sample was binned independently using three automatic binning programs: MetaBAT[96] (v2.12.1), CONCOCT[97] (v1.1.0), and MaxBin[98] (v2.2.7). DasTool[99] (v1.1.1) was then used to select the best bacterial bins from the combination of these three automatic binning programs. The resulting draft genome bins were dereplicated at 98% whole-genome average nucleotide identity (gANI) via dRep (v3.2.2)[100], using minimum completeness of 75%, maximum contamination of 10%, the ANImf algorithm, 98% secondary clustering threshold, and 25% minimum coverage overlap. Genomes with gANI ≥95% were classified as the same species, and the genome with the highest score (as determined by dRep) was chosen as the representative genome from each species.

## Taxonomy assignment

The amino acid sequences of predicted genes in all assembled bins were searched against the UniProt100 database using the *usearch* (v10.0.240) *ublast* with a maximum e-value of 0.0001. tRep (https://github.com/MrOlm/tRep/tree/master/bin) was used to convert identified taxonomic IDs into taxonomic levels. Briefly, for each taxonomic level (species, genus, phylum, etc.), a taxonomic label was assigned to a bin if >50% of proteins had the best hits to the same taxonomic label. GTDB-Tk[101] (v2.2.6) was used to resolve taxonomic levels that could not be assigned by tRep.

## Pairwise genomic comparison of *B. breve* from FT-1, FT-2, and PT-1

Genome-wide average nucleotide identity (gANI) for the three *B. breve* genomes reconstructed from FT-1, FT-2, and PT-1 were calculated using dRep *compare* with the ANIm algorithm.

## Genome metabolic annotation

Kyoto Encyclopedia of Genes and Genomes (KEGG) orthology groups (KOs) were assigned to predicted ORFs for all fecal metagenomes using KofamKOALA[102] (v1.3). Carbohydrate active enzymes (CAZymes) were assigned to all nucleotide sequences using run_dbcan.py (v4.0.0) (https://github.com/linnabrown/run_dbcan) against the dbCAN HMM (v11), DIAMOND (v2.0.9), and eCAMI with default settings (databases were downloaded following the instructions of run_dbcan). Final CAZyme domain annotations were the best hits based on the outputs of all three databases. Domains were also predicted using hmmsearch (v.3.3.2) (e-value cut-off $1 \times 10^{-6}$) against the Pfam r35 database[103]. The domain architecture of each protein sequence was resolved using cath-resolve-hits with default settings[104]. The transporters were predicted using hmmsearch (same settings as the Pfam prediction and domain architecture was resolved using *cath-resolve-hits* (v0.16.5)) and BLASTP (v2.12.0+) (keeping the best hit, e-value cutoff 1e-20) against the Transporter Classification Database (TCDB) (downloaded in October 2022)[105]. SignalP (v.5.0b) was used to predict proteins' putative cellular localization[106].

## Metagenomics prediction on *B. breve*'s 2'FL utilization

Fucosidases GH29 and GH95, as well as their cellular locations, were searched on annotated *B. breve* genomes (see Method section "Genome metabolic annotation"). To search for potential 2'FL transporters, we first examined ten genes upstream and downstream of the identified *B. breve* GH95 fucosidase. No transporter hits were identified using a combination of the TCDB and KEGG databases. We next searched for homologs of two well-characterized *Bifidobacterium* fucosyllactose transporters using BLASTP[28,107]. All de novo assembled contigs from enrichments were used to account for potential mis-binning of the reconstructed *B. breve* genomes. The closest hit shares ~44% amino acid identity with the reference transporters, and its neighboring

genes are not annotated as being involved in fucose- or lactose-related metabolism. We further mapped sequencing reads from enrichments against the reference fucosyllactose transporters and no hits were detected.

## Metagenomics prediction on D-lactose and L-fucose metabolism

To assess D-lactose and L-fucose metabolism, we first manually curated a list of relevant genes based on extensive literature searches[62,71,108–112] (Supplementary Data Files 1 and 2). Gene detection was primarily achieved using de novo assembled genomes (see "Metagenomic de novo binning" and "Genome metabolic annotation" above). When de novo genomes are unavailable (i.e., *Clostridium neonatale* from FT-1, which was present in <1% in all enrichments), data from read-mapping to the 1005 representative subspecies (see "Read-mapping-based species detection") were used and genes were profiled via inStrain *profile* under default settings. Specifically, genes were considered present if they had ≥1× coverage across ≥70% of their length. For an organism predicted to metabolize D-lactose, it must encode lactose transporter(s) and β-galactosidase(s). For L-fucose metabolism, organisms must encode fucose transporter(s) and all genes involved in one of the L-fucose metabolic pathways (as listed in[62,109]) to be identified as capable of L-fucose metabolism. Predicted pathways for each species were manually verified.

## Community diversity analysis

Modules from scikit-bio (v0.5.6) were used for the weighted ("skbio.diversity.beta.weighted_unifrac") and unweighted ("skbio.diversity.beta.unweighted_unifrac") UniFrac distances. The phylogenetic tree used in UniFrac distances was constructed by comparing all bacterial genomes grown in enrichments to each other using dRep *compare* with a mash sketch size of 10,000.

## Principal components analysis

Principal components analysis was performed using *scikit-learn* and was conducted based on the relative abundance of bacterial genomes in enrichments as assessed using the weighted UniFrac distance.

## Data analysis and plotting

All data analyses were done using python (v3.9.1). Data plots were generated using Matplotlib (v3.4.2) and/or Seaborn (v0.11.1). Figures 1a and 6a were generated using BioRender and publication licenses were obtained.

## Statistics and reproducibility

**Two-group univariate comparisons.** Statistical significance was calculated using a two-sided Wilcoxon rank-sum test (implemented using the SciPy (v1.6.3) module "scipy.stats.ranksums") or Welch's *t*-test (implemented using the SciPy module "scipy.stats.ttest_ind" with equal_var = False). Correlations were calculated using the Spearman correlation (implemented using the SciPy module "scipy.stats.spearmanr"). Multiple comparisons were false discovery rate (FDR) corrected using Benjamini–Hochberg correction with a threshold of $q < 0.05$.

**Two-sided linear least-squares regression.** For each infant stool inoculum, to calculate the reproducibility of the growth conditions, linear least-squares regression (implemented using the SciPy module "scipy.stats.linregress") was applied to calculate the $\log_{10}$(relative abundance) of each species among pairs of replicates for all combinations of inoculum and medium.

**Sample selection.** No statistical method was used to predetermine the sample size. Stool inocula were chosen solely based on microbiome compositions, which were all collected and analyzed in our

previous study[45]. Except for contaminants (see "Read-mapping-based species detection"), no data were excluded from the analyses. Randomization and blinding were not applicable to this study. Thus, the experiments were not randomized, and the investigators were not blinded to allocation during experiments and outcome assessment.

## Reporting summary

Further information on research design is available in the Nature Portfolio Reporting Summary linked to this article.

## Data availability

The metagenome-assembled genomes data generated in this study have been deposited on Figshare under the accession code https://doi.org/10.6084/m9.figshare.22320865 (https://figshare.com/articles/dataset/Lou_et_al_2023_bioRxiv_de_novo_bacterial_genomes/22320865). Reads of the infant stool inocula are available under SRA accessions SRS8184257 (FT-1; L2_031_090G1) (https://www.ncbi.nlm.nih.gov/biosample/?term=SRS8184257%20), SRS8184183 (FT-2; L3_130_056G1), and SRS8184427 (PT-1; L3_069_015G1). All data used for generating the main and supplementary figures are provided in the Source Data. The 1005 representative subspecies used for this study's read-mapping-based species detection were generated from our previous study[45] and are available on Figshare: https://doi.org/10.6084/m9.figshare.13667816 (https://figshare.com/articles/online_resource/dRepGenomes_Lou2021_tar_gz/13667816). The UniRef100, Pfam r35, NCBI BLASTP, and the Transporter Classification Database (TCDB) databases used for annotation in this study are publicly available (https://www.uniprot.org/help/uniref, http://pfam.xfam.org/, https://ftp.ncbi.nlm.nih.gov/blast/ and https://www.tcdb.org/, respectively). Source data are provided with this paper.

## Code availability

The software used in this paper is publicly available. No new code was generated.

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

## Acknowledgements

We thank Michael J. Morowitz and Brian A. Firek for collecting infant stool samples characterized in prior work[45]; Rohan Sachdeva, Jordan Hoff, and Shufei Lei for their technical support; Netravathi Krishnappa and the rest of the IGI sequencing core for sequencing support; Steven A. Frese and Alexander Crits–Christoph for experimental suggestions; Jacob West-Roberts for functional analyses support; Matthew R. Olm for comments on the manuscript. For funding support, we acknowledge NIH award RAI092531A (J.F.B. and M.J.M.).

## Author contributions

Y.C.L., B.E.R., A.L.B., J.A.D., and J.F.B. designed the study; Y.C.L., K.S.D., B.E.R., R.R., and L.S. conducted experiments; Y.C.L., K.S.D., B.E.R., and R.R. performed DNA extractions and metagenomics library preparations for all samples; M.C.S. assisted with functional enrichment analyses; Y.C.L. performed data analyses; Y.C.L. and J.F.B. wrote the paper; all authors contributed to the paper revision.

## Competing interests

The Regents of the University of California have a patent pending related to this work on which Y.C.L., B.E.R., A.L.B., and J.F.B. are inventors. J.F.B. is a co-founder of Metagenomi. J.A.D. is a co-founder of Caribou Biosciences, Editas Medicine, Scribe Therapeutics, Intellia Therapeutics, and Mammoth Biosciences. J.A.D. is a scientific advisory board member of Vertex, Caribou Biosciences, Intellia Therapeutics, Scribe Therapeutics, Mammoth Biosciences, The Column Group, and Inari. J.A.D. is a Director at Johnson & Johnson, Altos Labs, and Tempus and has research projects sponsored by AppleTree Partners and Roche. B.E.R. is a shareholder of Caribou Biosciences, Intellia Therapeutics, Locus Biosciences, Inari, TreeCo, and Ancilia Biosciences. The authors declare no competing interests.
