## [Peer Review File · Nature Communications]

REVIEWER COMMENTS

Reviewer #2 (Remarks to the Author):

The authors have not addressed most of my questions, so my scientific comments remain essentially the same as the first review. I find the work generally interesting and well done, and I appreciate the editing changes to the text to address my comments related to presentation.

Reviewer #3 (Remarks to the Author):

Re-review of 2'FL manuscript

Thank you for the opportunity to review this revised manuscript, which we originally quite liked. While not many changes were made (and they are not easy to identify because no tracked-changes version was provide), we do acknowledge some improvements. We would like to see our original comments addressed more fully, and feel this is feasible given that we have only requested interpretive/textual changes (not any new experiments). Specifically:

1. New schematic in the Discussion section: we appreciate this addition but had expected the schematic would illustrate proposed mechanisms of interactions between bacteria (i.e. clearly including *R. gnavus* and *B. breve*, and demonstrating the digestion steps of 2'FL completed by each). Could this be changed, or added as a second panel? If the current figure is retained, we suggest featuring breast milk with and without 2'FL rather than formula.

2. We would like the authors to fully address our previous comment about secretor status— at a minimum, we would like to see the lack of data on secretor status to be discussed as a limitation to the current study. It's hard to assess without a tracked changes version of the manuscript, but we

believe that the only mention of secretor status added in this revised version is in the Introduction, where they note that 2'FL is the most abundant HMO in the milk of secretors. We would like to see the authors discuss their findings in light of non-secretors, who do not have 2'FL in their milk. What future studies could be done to expand on the current manuscript and assess patterns in non-secretors?

3. We would like the authors to fully address our previous comment about *B. infantis*—specifically, we asked the authors to discuss how their findings of cooperation between *B. breve* and *R. gnavus* expand our current understanding of *B. infantis*. As we previously asked, why not just administer *B. infantis* instead?

4. We disagree with the authors assertion that breastfeeding history and duration are unnecessary data for the current study. Feeding practices at and prior to the time of sample collection are both relevant to microbiome composition and function. Given the ecological nature of this paper, we believe that the authors should consider the potential impact of early life feeding (i.e. priority effects and related processes of community succession). At minimum, the lack of this information should be mentioned as a limitation to the current study.

5. While the authors included a definition for keystone species, it was added on page 14, not at the first mention of the term on page 10. This needs to be addressed for logical flow and clear reading of the manuscript.

6. Related, we would like to reiterate that the authors should consider including an ecological perspective when discussing precision medicine at the end of the Discussion. Since this was a major theme of the manuscript, it would be nice to see this perspective in the concluding sentences. How should the ecological findings of this paper inform precision medicine moving forward?

7. It would be more convenient for readers to have the supplemental tables combined into one spreadsheet with multiple tabs.

Reviewer #5 (Remarks to the Author):

Alternate for Ref 1.

Maj Comment 1. The current manuscript demonstrates that extracellular fucosidase from *R. gnavus* allows growth of *B. breve* on 2-FL which cannot utilize 2-FL alone but is cross-feed lactose by Rg by this activity. Ref 1 had cited both a paper that suggest *B. breve* is not impacted by cross-feeding in an infant study of 2-FL and a paper that reports that *B. breve* does have fucosidase activity and actually crossfeeds other fucosidase+ bacteria such as *Blautia* and Rg. Ref 1 had asked the authors to appease these differences and analyze the metagenomic data from these papers. The authors have done this to the extent that they are able, and their conclusion is basically that these two papers are limited in their conclusions that are contradictory to the present manuscript. Therefore the authors have decided not to cite these two paper. Perhaps the authors do not want to cite these so as not to appear as negatively criticizing these works. But I think it is glaring not to cite them and this can be done in a balanced way. The reader should have these two studies acknowledged so that they can weigh the seemingly contradictory findings themselves. Related to this, the authors should emphasize in there manuscript that clearly some Bbreve have fucosidase activity and some do not.

Maj Comment 2. Ref 1 asks the authors to cite 2 16S based studies. The authors do.

Maj Comment 3. Referring to Fig 7, Ref 1 asks the authors to knockout the GH95 fucosidase from *R. gnavus*. This is very technically challenging and in this alternative referee's opinion not necessary and a bit overkill.

Minor comments are addressed appropriately.

Point-by-point responses to Reviewers' comments

Below are our point-by-point responses to the Reviewers' comments, with the comments in *italics* and our response in regular text. The modifications are highlighted in the main text of the revised manuscript.

Reviewers' Comments:

Reviewer #2 (Remarks to the Author):

The authors have not addressed most of my questions, so my scientific comments remain essentially the same as the first review. I find the work generally interesting and well done, and I appreciate the editing changes to the text to address my comments related to presentation.

We are glad to hear that the reviewer finds our work generally interesting and well done. No additional revision was done in regard to this reviewer's experiment-related comments because we believe that our existing data robustly support the conclusions and that the suggested experiments, while potentially informative, may not substantially alter the essence of our findings.

Reviewer #3 (Remarks to the Author):

Re-review of 2'FL manuscript

Thank you for the opportunity to review this revised manuscript, which we originally quite liked. While not many changes were made (and they are not easy to identify because no tracked-changes version was provide), we do acknowledge some improvements. We would like to see our original comments addressed more fully, and feel this is feasible given that we have only requested interpretive/textual changes (not any new experiments).

We are glad to hear that the reviewer quite liked our work and appreciate all the detailed comments the reviewer brought up. We do want to emphasize that we tried our best to address all comments during our initial revision. We apologize for not highlighting all the changes, which we admit made the re-review of the manuscript more challenging.

In this newly revised version, we further developed our arguments in response to the reviewer's comment and have highlighted all the changes in the revised manuscript.

Specifically:

1. New schematic in the Discussion section: we appreciate this addition but had expected the schematic would illustrate proposed mechanisms of interactions between bacteria (i.e. clearly including R. gnavus and B. breve, and demonstrating the digestion steps of 2'FL completed by each). Could this be

changed, or added as a second panel? If the current figure is retained, we suggest featuring breast milk with and without 2'FL rather than formula.

Thank you for this very useful comment. We agree the suggestion of the diagram was excellent and we have now revised the figure by adding the names *R. gnavus* and *B. breve* and increasing the sizes of the 2'FL, L-fucose, and D-lactose molecules to make the interactions required for 2'FL metabolism more apparent. We also changed the bottle to a milk droplet. However, we do not diagram the case of +/- 2'FL because the paper is about how 2'FL influences the growth of *B. breve* in the presence/absence of extracellular-fucosidase-encoding organisms (thus, the droplets are the same, but the microbiomes differ). We also made the *B. breve* more numerous in the panel on the right in the case where breakdown products generated by *R. gnavus* were available. We hope these changes are acceptable.

*2. We would like the authors to fully address our previous comment about secretor status— at a minimum, we would like to see the lack of data on secretor status to be discussed as a limitation to the current study. It's hard to assess without a tracked changes version of the manuscript, but we believe that the only mention of secretor status added in this revised version is in the Introduction, where they note that 2'FL is the most abundant HMO in the milk of secretors. We would like to see the authors discuss their findings in light of **non-secretors**, who do not have 2'FL in their milk. **What future studies could be done to expand on the current manuscript and assess patterns in non-secretors?***

We thank the reviewer for this suggestion. In the re-revised manuscript, we added the following to the discussion (lines 660-665 on page 10 in the re-revised manuscript):

“In addition to the gut microbiome, other clinical factors such as maternal secretor status, diets and the immune system could affect the efficacy of microbiome interventions⁷⁴. Compared to secretors, non-secretor mothers lack the allele to produce 2'FL and other α 1-2 fucosylated oligosaccharides⁸⁵. In our study, maternal and infant secretor statuses were not collected. In future studies, it would be interesting to investigate how maternal and infant secretor statuses influence the outcome of microbiome interventions”

*3. We would like the authors to fully address our previous comment about *B. infantis*—specifically, we asked the authors to discuss how their findings of cooperation between *B. breve* and *R. gnavus* expand our current understanding of *B. infantis*. As we previously asked, why not just administer *B. infantis* instead?*

We thank the reviewer for this comment and agree that adding this component will make our manuscript more relevant to a wider range of audience. We did address it in our first revision and again apologize for not highlighting our changes. In the re-revised manuscript, we highlight these changes and make minor additional changes. To address how our work can expand our current understanding of *B. infantis*, we now state (lines 623-635 on pages 9-10 in the re-revised

manuscript):

“Besides clinical factors such as frequent antibiotic exposures and lack of sufficient breastfeeding^{68,69}, microbiome compositions and functions, such as those dominated by Proteobacteria, could also negatively affect the colonization of *Bifidobacterium*, especially those with a limited HMO metabolic capacity, including *B. breve*. Supplementation of *Bifidobacterium* probiotic strains with a broad HMO metabolic capacity, such as *B. longum* subsp. *infantis*, thus has been used to rescue dysbiotic preterm gut microbiomes^{57,59,60,65,70}. However, the efficacy of exogenously administered *Bifidobacterium* strains can be affected by the resident microbiome^{3,30,71}. For instance, despite having a limited HMO-metabolism ability, *B. breve* can outcompete other *Bifidobacterium* species, including *B. infantis*, if it colonizes the infant gut first⁷¹. By enriching indigenous *B. breve* using a targeted combination of 2'FL and *Ruminococcus gnavus*, our study provided an alternative to probiotic administration”

4. We disagree with the authors assertion that breastfeeding history and duration are unnecessary data for the current study. Feeding practices at and prior to the time of sample collection are both relevant to microbiome composition and function. Given the ecological nature of this paper, we believe that the authors should consider the potential impact of early life feeding (i.e. priority effects and related processes of community succession). At minimum, the lack of this information should be mentioned as a limitation to the current study.

We agree with the reviewer that the breastfeeding history and duration (prior to collection of the sample used for inoculation) are important for the current study. We stated in our original response: “we only provided information on feeding practice prior to sample collection as it can affect the gut microbiome composition of the samples we cultivated.” As all of the research involved microbiome changes in our enrichments, the subsequent feeding history of the infants after the point that the sample used for inoculation was collected is not relevant to this study. There is no record of any infant drinking formula and all infants were drinking breast milk only by the time the sample was collected. We chose to use samples from infants only fed breast milk, as stated in the first revised version of our manuscript.

5. While the authors included a definition for keystone species, it was added on page 14, not at the first mention of the term on page 10. This needs to be addressed for logical flow and clear reading of the manuscript.

We thank the reviewer for this feedback. In the re-revised manuscript, we have moved the definition of the keystone species to page 10 in the re-revised manuscript (lines 388-389 on page 7). We have also expanded the discussion, as follows (lines 637-645 on pages 10 in the re-revised manuscript):

“*R. gnavus* is prevalent in both healthy infant and adult gut microbiomes and their capability in metabolizing complex carbohydrates, including mucin and HMOs, partly contribute to their stable gut colonization since early-life⁷². In our study, microbiome cultivation using

2'FL revealed *R. gnavus* as a keystone species. Keystone species have a disproportionately large importance in ecosystems relative to their abundance^{73,74}. *R. gnavus*'s disproportionately large impact on some microbial communities thus makes it a potential ideal candidate for microbiome intervention. However, additional work is needed for identifying specific *R. gnavus* probiotic strains as some members of this species were found to be strongly associated with diseases including inflammatory bowel diseases (IBD)^{72,75}."

6. *Related, we would like to reiterate that the authors should consider including an ecological perspective when discussing precision medicine at the end of the Discussion. Since this was a major theme of the manuscript, it would be nice to see this perspective in the concluding sentences. How should the ecological findings of this paper inform **precision medicine** moving forward?*

We have now added a final paragraph to the Discussion that describes how the findings from our work can inform precision medicine (lines 667-704 on pages 10-11 in the re-revised manuscript).

"In summary, by cultivating different infant gut microbiomes, we showed that community compositions influence HMO metabolism. The microbiome cultivation systems established here can be extended to experimentally elucidate individualized responses to other perturbations, such as antibiotics and other complex carbohydrates. The model microbiomes developed in this work can also be used in microbiome editing-based experiments to study mechanisms of inter-organism interactions in communities. For instance, we recently developed a CRISPR-Cas-based approach to editing organisms in the context of their microbiomes and demonstrated strain-specific editing sensitivity in an *in vitro* infant gut model⁸⁶.

The gut microbiome interacts with supplemented compounds bidirectionally, in so far as the compounds impact the microbiome composition and the microbiome members impact the compounds⁸⁷. Significant impacts of these interactions on our health can arise through microbiome processing of dietary compounds, alteration of the efficacy of pre- and probiotics, and moderation of the impact of prescription medications⁸⁷. As each person is colonized by a distinct gut microbiome, it may be possible to maximize the effectiveness of an intervention or treatment by evaluating a compound's impact on the individual's microbiome in a laboratory microbiome experiment. Overall, our findings demonstrate a way in which gut microbiome experiments can be an integral part of precision medicine. Insights gained from these individualized *in vitro* microbiomes could enable more precise and effective treatments."

7. *It would be more convenient for readers to have the supplemental tables combined into one spreadsheet with multiple tabs.*

We have combined all our supplemental figures into one table with multiple tabs.

Reviewer #5 (Remarks to the Author):

Alternate for Ref 1.

Maj Comment 1. The current manuscript demonstrates that extracellular fucosidase from R. gnavus allows growth of B. breve on 2-FL which cannot utilize 2-FL alone but is cross-feed lactose by Rg by this activity. Ref 1 had cited both a paper that suggest B. breve is not impacted by cross-feeding in an infant study of 2-FL and a paper that reports that B. breve does have fucosidase activity and actually crossfeeds other fucosidase+ bacteria such as Blautia and Rg. Ref 1 had asked the authors to appease these differences and analyze the metagenomic data from these papers. The authors have done this to the extent that they are able, and their conclusion is basically that these two papers are limited in their conclusions that are contradictory to the present manuscript. Therefore the authors have decided not to cite these two paper. Perhaps the authors do not want to cite these so as not to appear as negatively criticizing these works. But I think it is glaring not to cite them and this can be done in a balanced way. The reader should have these two studies acknowledged so that they can weigh the seemingly contradictory findings themselves.

We believe the reviewer misunderstood our response. As we stated in our response, we “have cited both studies in our revised manuscript.” We did not include any of our metagenomics analyses using data from these two papers as “the findings do not inform the current study.”

Specifically, we do not think findings from these two studies were contradictory to ours. For the study conducted by Wallingford and colleagues, we did not see a positive correlation between 2'FL concentration and the growth of *B. breve* and we reasoned that this may be due to the fact that only 0.1% of 2'FL was added. As shown in our data, *B. breve* demonstrated significant growth only when $\geq 0.15\%$ of 2'FL was added. Wallingford and colleagues also acknowledged in their study that the amount of administered 2'FL might be too low. For the second study, *B. breve* was found only at low abundance and was not reported in cultivated microbiomes. Since our study specifically focused on how 2'FL may influence the growth of *B. breve* in gut microbiomes, we did not include the data from this study in our manuscript. However, we did cite both studies in our revised manuscript (ref numbers 52 and 53).

Related to this, the authors should emphasize in there manuscript that clearly some Bbreve have fucosidase activity and some do not.

We have now emphasized this point in our introduction (see the bolded text below; lines 67-70 on page 2 in the re-revised manuscript).

Further, variations among strains of the same *Bifidobacterium* species also contribute to interpersonal differences in 2'FL utilization. For instance, only a minority of strains of *Bifidobacterium breve*, an abundant infant gut colonizer, can break down 2'FL **by encoding both the α -L-fucosidase and 2'FL transporter**^{12,19-21}.

Maj Comment 2. Ref 1 asks the authors to cite 2 16S based studies. The authors do.

We thank the reviewer for acknowledging our revision!

Maj Comment 3. Referring to Fig 7, Ref 1 asks the authors to knockout the GH95 fucosidase from R. gnavus. This is very technically challenging and in this alternative referee's opinion not necessary and a bit overkill.

We thank the reviewer for agreeing with us that the experiment suggested by reviewer #1 was not necessary!

Minor comments are addressed appropriately.

We thank the reviewer for acknowledging our revision!